# High-Frequency Imaging Reveals Synchronised Delta- and Theta-Band Ca^2+^ Oscillations in the Astrocytic Soma In Vivo

**DOI:** 10.3390/ijms25168911

**Published:** 2024-08-16

**Authors:** Márton Péter, László Héja

**Affiliations:** 1Institute of Organic Chemistry, HUN-REN Research Centre for Natural Sciences, Magyar tudósok körútja 2, 1117 Budapest, Hungary; peter.marton@ttk.hu; 2Hevesy György PhD School of Chemistry, ELTE Eötvös Loránd University, 1117 Budapest, Hungary

**Keywords:** astrocytes, fast Ca^2+^ signals, Ca^2+^ oscillations, slow-wave sleep, high-frequency imaging, astrocyte–neuron interactions, astrocyte network

## Abstract

One of the major breakthroughs of neurobiology was the identification of distinct ranges of oscillatory activity in the neuronal network that were found to be responsible for specific biological functions, both physiological and pathological in nature. Astrocytes, physically coupled by gap junctions and possessing the ability to simultaneously modulate the functions of a large number of surrounding synapses, are perfectly positioned to introduce synchronised oscillatory activity into the neural network. However, astrocytic somatic calcium signalling has not been investigated to date in the frequency ranges of common neuronal oscillations, since astrocytes are generally considered to be slow responders in terms of Ca^2+^ signalling. Using high-frequency two-photon imaging, we reveal fast Ca^2+^ oscillations in the soma of astrocytes in the delta (0.5–4 Hz) and theta (4–8 Hz) frequency bands in vivo in the rat cortex under ketamine–xylazine anaesthesia, which is known to induce permanent slow-wave sleep. The high-frequency astrocytic Ca^2+^ signals were not observed under fentanyl anaesthesia, excluding the possibility that the signals were introduced by motion artefacts. We also demonstrate that these fast astrocytic Ca^2+^ signals, previously considered to be exclusive to neurons, are present in a large number of astrocytes and are phase synchronised at the astrocytic network level. We foresee that the disclosure of these high-frequency astrocytic signals may help with understanding the appearance of synchronised oscillatory signals and may open up new avenues of treatment for neurological conditions characterised by altered neuronal oscillations.

## 1. Introduction

Astrocytes have become a new focus of neuroscience in recent decades. The homeostatic role of these cells has been firmly established since their discovery, including the metabolic maintenance of neurons, regulation of the extracellular medium, contributions to the immune response, and the formation of the blood–brain barrier. Several key discoveries, however, have prompted a paradigm shift in our understanding of astrocytes. In addition to their structural and supporting roles, these cells have been shown to be involved in the modulation of neuronal functions in a multitude of ways [1]. Astrocytes are now considered to be integral elements of the tripartite synapse, exchanging information with both pre- and postsynaptic neurons [2]. A single astrocyte may envelop hundreds of thousands of synapses [3], thereby influencing the neuronal function at a network level. In addition, astrocytes form their own network through gap junction connections and have been demonstrated to propagate large-scale synchronised Ca^2+^ signals through these connections [4], which makes them perfectly suitable to initiate or maintain synchronised activity in large-scale networks [5,6,7,8].

The identification of network-level synchronised oscillatory neuronal activity at different frequency bands and its association with specific physiological and pathological processes was a major milestone in the development of neurobiology [9,10]. However, despite the ability of astrocytes to contribute to these synchronised oscillations, investigations of astrocytic activity at sampling frequencies higher than 1 Hz are largely missing, despite the recent advances in fast calcium imaging techniques that currently allow for the detection of changes in intracellular Ca^2+^ at up to 160 Hz in a single plane [11] or even 40 Hz in a 3D volume [12]. Astrocytes are generally considered to be slow responders, with somatic Ca^2+^ spikes operating on the second scale [4,13]. Consequently, somatic astrocytic Ca^2+^ signals are generally measured at sampling frequencies of less than 2 Hz [4,13,14,15,16,17], which does not allow for the detection of signals at >1 Hz. Although somatic Ca^2+^ signals originate from distinct astrocytic microdomains that are capable of producing rapid Ca^2+^ signals on a similar time scale to neuronal activity [14,18,19,20,21,22,23,24,25], the integrated signals on the soma that can be synchronised through the astrocytic network [26] are usually considered to be discrete spikes, i.e., single events, instead of high-frequency, recurring, oscillatory signals.

Here, we present evidence for the appearance of synchronised high-frequency somatic Ca^2+^ signals, as ascertained using high-frequency 2P microscopy in vivo in rats during ketamine–xylazine anaesthesia. We also demonstrate that the observed astrocytic Ca^2+^ signals in the delta and theta frequency bands are present in a large population of astrocytes and are synchronised in the astrocytic network. These data suggest that astrocytic Ca^2+^ signals have a wider range of complexity and more rapid kinetics than previously thought.

## 2. Results

### 2.1. Astrocytes Display High-Frequency Ca^2+^ Oscillations

We simultaneously monitored astrocytic and neuronal activity in the visual cortex of rats in vivo using Oregon Green BAPTA (OGB-1) fluorescence. Astrocytic loading of OGB-1 was confirmed by labelling with the astrocyte-specific [27] sulforhodamine-101 (SR101). Both dyes were delivered to the cortical surface through a cannula before the rats were anaesthetised with ketamine–xylazine. Imaging sessions started 60 min after anaesthesia. OGB-1 fluorescence was monitored in 14–18 consecutive 60 s long imaging sessions. In each of the 50 imaging sessions in three animals, 46–148 cells (103 ± 3 cells on average) in the field of view (a 446 × 446 or 602 × 602 µm area) were recorded. Of the OGB-1 loaded cells, 90 ± 2% were identified as astrocytes through colocalisation with SR-101 fluorescence. To enable the detection of fast Ca^2+^ signals, OGB-1 fluorescence dynamics were recorded in high-speed line scan mode. Two crossing scan lines were automatically applied to each identified cell using custom MATLAB scripts. The scanning order of the cells was optimised using the travelling salesman method. Using these techniques, 46–215 Hz sampling frequencies could be achieved.

Traditionally, astrocytic Ca^2+^ signals are analysed by detecting individual spikes on the normalised ΔF/F_0_ traces. These events correspond to substantial but slow changes in the somatic Ca^2+^. Using high-frequency imaging, we were able to detect much faster somatic Ca^2+^ fluctuations. These were characteristically oscillatory signals that could be observed in both neuronal and astrocytic soma (Figure 1B,C) in vivo under ketamine–xylazine anaesthesia. To analyse these signals, we applied spectral analysis to the recorded traces, rather than characterising the individual spikes.

By investigating the frequency bands of the somatic Ca^2+^ signals in astrocytes and neurons, we identified activity in two distinct ranges: the delta (0.5–4 Hz) and the theta (4–8 Hz) bands (Figure 1D,E). Wavelet analysis revealed that a strong frequency component between 1 and 2 Hz is present in both neurons and astrocytes (Figure 1D,E). This slow-wave activity signal is characteristic for non-REM sleep phases, which are known to be induced by ketamine–xylazine anaesthesia [28,29]. In addition, many cells also showed oscillatory activity between 2 and 4 Hz in the high delta range. Wavelet analysis also revealed a theta-range oscillatory activity between 4 and 8 Hz. (Figure 1D,E). Importantly, both delta- and theta-range activities were observed in both astrocytes and neurons.

Networking cells are gradually recruited into oscillatory activity [5]. Since we started imaging sessions 60 min after anaesthesia, most of the identified cells were already involved in the delta- and theta-range network oscillations. However, we could also identify both neurons and astrocytes that did not display any significant oscillatory activity (Appendix A), confirming that the observed oscillatory signals were not results of motion artefacts.

### 2.2. High-Frequency Ca^2+^ Signals Do Not Emerge in Fentanyl Anaesthesia

The appearance of delta- and theta-range activity in astrocytes and neurons is most likely induced by ketamine–xylazine anaesthesia, which generates permanent slow-wave sleep [28,29]. To further confirm that the observed high-frequency astrocytic signals corresponded to the induced slow-wave activity and were not caused by motion artefacts, we investigated the appearance of high-frequency Ca^2+^ signals in three rats anaesthetised by fentanyl (25 µg/kg), medetomidine (0.25 mg/kg), and midazolam (2 mg/kg). In these 84 imaging sessions, 7–70 cells (28 ± 2 cells in average) were recorded at 29–101 Hz sampling frequencies, and 95 ± 1% of them were identified as astrocytes. In fentanyl anaesthesia, most cells did not show delta- or theta-band Ca^2+^ oscillations (Figure 2), confirming that the high-frequency astrocytic Ca^2+^ signals observed during ketamine–xylazine anaesthesia were not the results of motion artefacts triggered by breathing, heartbeat, or movement instability of the scanning head.

### 2.3. High-Frequency Astrocytic Ca^2+^ Signals Are Present at the Network Level

After detecting oscillatory activity in single astrocytes and neurons, we investigated whether this activity appeared only at the level of single cells or if it was widespread at the network level in ketamine–xylazine anaesthesia. Since we simultaneously monitored Ca^2+^ signals in 46–148 cells in each measurement, we were able to determine whether delta- and theta-band activity could be simultaneously observed in a large number of networked cells. To this end, we first applied wavelet analysis to all identified cells. Then, the wavelet transform was summed along the time axis at each frequency for the whole duration of the recording. Finally, the frequency distribution was displayed for all astrocytes and neurons within an imaging session (Figure 3A,C). Data from two representative imaging sessions demonstrated that delta- and theta-band activity was present in the majority of both cell types, revealing that these oscillatory activities were synchronised at the network level (Figure 3). A characteristic peak in the frequency domain at 1–2 Hz was observed in both astrocytes and neurons (Figure 3A,C). This delta-band activity was accompanied by another peak in the theta range at 6–7 Hz (Figure 3A) or around 5 Hz (Figure 3C). The delta-band activity was highly synchronised between all participating cells, while the theta-band activity showed a slight variance in frequency, especially in astrocytes (Figure 3A). The network-level synchronisation was also confirmed by spectral analysis of the simultaneously measured local field potential, which also showed increased power at the frequencies identified by calcium imaging (Figure 3B,D).

### 2.4. Delta- and Theta-Band Oscillations in Astrocytes Are Synchronised in the Network

After demonstrating that delta- and theta-band oscillations were present across a large population of astrocytes, we assessed whether this activity was synchronised in the network or if they spontaneously appeared in individual cells. To investigate this issue, we calculated the phase-locking value (PLV) to quantify synchronisation between all cell pairs under both ketamine–xylazine (N = 298,838 cell pairs from three animals) and fentanyl (N = 82,682 cell pairs from three animals) anaesthesia. Phase synchronisation results for representative imaging sessions indicated that highly phase-synchronised activity was widespread in both the delta (Figure 4A) and the theta bands (Figure 4C) in ketamine–xylazine anaesthesia, but not in fentanyl anaesthesia (Figure 4A,C). To quantify the connections in and between the astrocytic and neuronal networks, we calculated PLV separately for astrocyte pairs, neuron pairs, and astrocyte–neuron pairs. Phase-coupling in all three groups was stable, and it was significantly higher in ketamine–xylazine anaesthesia compared to fentanyl anaesthesia both in the delta (Figure 4B) and theta bands (Figure 4D).

The distribution of PLV in all animals confirmed that significantly more astrocyte–astrocyte, neuron–neuron, and astrocyte–neuron cell pairs showed high (>0.4) phase-coupling during ketamine–xylazine anaesthesia (Figure 5A,C). Furthermore, the average PLV was found to be higher in the neuronal network compared to the astrocytic network in ketamine–xylazine anaesthesia, while lower phase-coupling was observed between neurons than between astrocytes in animals anaesthetised with fentanyl (Figure 5B,D). The average PLV was found to be significantly (*p* < 0.001) higher in all cell types under ketamine–xylazine anaesthesia compared to fentanyl anaesthesia for both delta- and theta-band signals.

## 3. Discussion

In this article, we present new evidence for high-frequency Ca^2+^ activity in the soma of astrocytes in the rat visual cortex under ketamine–xylazine anaesthesia. Ketamine–xylazine anaesthesia is known to induce permanent slow-wave activity in the neuronal network [28,29], which is often accompanied by theta oscillations [30,31]. We have demonstrated that astrocytes display oscillatory calcium signals in both the delta (0.5–4 Hz) and theta (4–8 Hz) ranges. All signals were present and synchronised at the network level in both astrocytes and neurons in ketamine–xylazine anaesthesia, but not in fentanyl anaesthesia, which does not induce slow-wave activity, excluding motion artefacts as the source of the observed high-frequency astrocytic signals. Both the delta- and theta-band activities were found to be highly phase-coupled, demonstrating that they are not independently generated in each astrocyte, but rather represent a coherent synchronised signal in the astrocytic network. To our knowledge, this is the first report of synchronised astrocytic somatic Ca^2+^ signals above 2 Hz.

Although our current goal was merely to reveal the existence of high-frequency oscillations in astrocytic soma, the appearance and widespread presence of delta- and theta-band activities in astrocytes raises the question of whether these oscillations originate from the astrocytes themselves or if they are mirroring the already established activity patterns of the neuronal network. Autonomous signals within the astrocytic syncytium can arise as a result of several mechanisms and sources. Astrocytic microdomains are capable of generating autonomic Ca^2+^ oscillations via the interplay between Na^+^-dependent transporters, such as GAT3 [32] or EAAT2 [33], and NCX. The influx of Na^+^ can reverse the operation of NCX, thereby resulting in the uptake of Ca^2+^ [34]. This has been confirmed by our previous in silico simulations [24]. In the astrocytic soma and main processes, the release of Ca^2+^ from intracellular stores can contribute to the development of autonomous Ca^2+^ signals, through the activation of GPCRs (like InsP_3_R2) and the subsequent activation of IP_3_-dependent release mechanisms [22,35]. Even in the absence of these mechanisms, spontaneous Ca^2+^ signals can still arise as a result of mitochondrial activity, specifically through the activation of the permeability transition pore (mPTP) [22,35]. The depletion of Ca^2+^ stores can also trigger Ca^2+^ entry trough STIMs and Orai channels [35,36].

Alternatively, the signals may propagate from the synapses, passed on to astrocytic microdomains, using extracellular Ca^2+^ sources [20,21]. If the synchronised neuronal activity is strong enough to activate a significant number of astrocytic microdomains, the signal could spread to the soma through the main processes, where intracellular Ca^2+^ stores are used for further propagation [14,26,37]. From here, the signals can spread through the astrocytic syncytium through gap junctions, explaining the network-wide activity in this cell population. Previous experimental data, however, suggest that Ca^2+^ signalling between neurons and astrocytes may not be unidirectional. Under similar circumstances, synchronised slow-wave activity of astrocytes developed earlier than that of neurons [5]. Furthermore, the specific blockade of astrocytic gap junctions resulted in a significant reduction in neuronal synchronisation in the slow-wave-activity frequency range (0.5–2 Hz) [5] and also diminished high-frequency epileptic activity [38]. In vitro data also demonstrate that specific inhibition or enhancement of astrocytic Ca^2+^ activity can decrease or increase neuronal synchronisation, respectively [13]. Further studies will be required to discriminate between neuronal and inherent astrocytic activity as the source of the fast astrocytic oscillations.

It is important to note that both delta- and theta-band oscillations are thought to play major roles in memory formation [39,40]. Our results, combined with previous observations about the causal role of astrocytes in generating slow-wave activity [5,6,8,41] and long-term potentiation [42,43], collectively establish the foundation for the therapeutic potential of astrocytes in the treatment of diseases associated with memory loss, like Alzheimer’s disease [44,45,46].

## 4. Materials and Methods

### 4.1. Animals

All animal procedures were performed according to standard ethical guidelines and approved by the local Animal Care Committee and the Government Office for Pest County (reference numbers PEI/001/3671-4/2015 and 22.1/2727/3/2011). Adult male and female Wistar rats (ToxiCoop, Budapest, Hungary) of at least 300 g were used throughout this study. Animals were housed in a 12 h light/dark cycle with food and water available ad libitum. Testing occurred in the light phase.

### 4.2. Surgical Preparation

All animals were familiarised with the experimenter on a daily basis over the week before surgery. Surgery was performed under ketamine–xylazine anaesthesia (100 mg/kg ketamine, Alpha-Vet, Székesfehérvár, Hungary; 10 mg/kg xylazine–hydrochloride, Sigma-Aldrich, St. Louis, MO, USA), along with 0.01 mg/kg of ropivacaine–hydrochloride (Naropin, AstraZeneca, Budapest, Hungary) as a local anaesthetic. The depth of anaesthesia was assessed by monitoring the pinch reflex of the animals. An ophthalmic ointment (Opticorn A, Cardon Pharmaceutical, Brugge, Belgium) was applied to the eyes to prevent corneal dehydration during surgery.

All animals were fitted with a cannula through a hole drilled into the skull, extending into the epidural space for dye injection. The hole was drilled right on the frontal edge of the coronal suture, 1.8 mm laterally from the sagittal suture on the right side of the skull. The cannula had to be inserted at a 45° frontal angle to accommodate the microscope objective during measurements. A circular cranial window with 5 mm diameter was implanted over the cortex, substituting the middle part of the right parietal bone. The dura mater was carefully removed from the part of the cortex covered by the window.

The steel legs of a JST-XH Male 2 Pin Connector were used as electrodes. The legs were spread 7.5 mmapart from each other and secured into holes drilled through the skull. Due to its size, the connector had to be fitted above the contralateral hemisphere from the cranial window to allow enough space for the microscope objective during measurements. The first hole was drilled at the edge of the coronal suture through the parietal bone on the very edge of the exposed part of the skull, the second one 7.5 mmdorsally, towards the middle of the parietal bone.

Finally, one part of an elongated rectangular aluminium plate was fixed to the left posterior quadrant of the exposed skull to enable head fixation during measurements.

All implanted objects were fixed in place by covering the exposed parts of the skull with Super-Bond dental cement (Sun Medical, Moriyama, Japan). Recuperating animals were treated daily with subcutaneous injections of 1 mg/kg of meloxicam (Medicus Partner Kft, Biatorbágy, Hungary) for at least two days after surgery. The cannula and electrodes were manufactured in-house using commercially available parts. The animals were kept in individual cages and were allowed to rest for one week before the start of testing.

### 4.3. Two-Photon Microscopy Coupled with Field Potential Recording

At 150 min prior to imaging sessions, animals were injected with 160 µM OGB-1 and 140 µM SR-101, suspended in 10 µL ACSF, through the implanted epidural cannula. Measurements were performed either under ketamine–xylazine anaesthesia, as used during surgical preparation (supplemented occasionally as needed during the measurements), or under fentanyl anaesthesia (fentanyl 25 µg/kg, Hungaropharma, Budapest, Hungary; medetomidine 0.25 mg/kg, Alpha-Vet, Székesfehérvár, Hungary; midazolam 2 mg/kg, Hungaropharma, Budapest, Hungary). Fentanyl anaesthesia was reversed after the experiments with a combination of antagonists (naloxone 0.6 mg/kg, Hungaropharma, Budapest, Hungary; atipamezole hydrochloride 1.25 mg/kg, Alpha-Vet, Székesfehérvár, Hungary; flumazenil 0.25 mg/kg, Hungaropharma, Budapest, Hungary). Head fixation was achieved using the implanted aluminium plate or a stereotaxic frame. With the head fixed, the rest of the body was rotated so the animal was laying on its left side. This helped to minimise any movement artifacts resulting from breathing during imaging.

Signals from the pair of implanted electrodes were recorded with a Multiclamp 700A amplifier (Axon Instruments, Foster City, CA, USA) and sampled at 10 kHz (Digidata 1320 A, Axon Instruments). Imaging data were recorded from 100 to 300 µm depth, corresponding to layer 2 of the V1 primary visual cortex, using a Femto2D-Dual two-photon microscope (Femtonics, Budapest, Hungary), equipped with a 10× water immersion objective. Cells were excited by a 920 nm laser (FemtoFiber ultra 920, Toptica Photonics AG, Graefelfing, Germany). OGB-1 fluorescence was detected at 475–575 nm, and SR-101 at 600–700 nm. We imaged 7–148 cells simultaneously in 60 s long line-scan sessions at 29–215 Hz. Data acquisition was performed using semi-automatic MATLAB scripts under manual supervision. After acquiring a 512 × 512-pixel image of the field of view, cells were identified on the OGB-1 channel based on their fluorescence intensity and size. Changes in OGB-1 fluorescence were detected in line scanning mode to achieve a high sampling frequency. The scanning path was determined automatically based on the position of identified cells. Importantly, two crossing line paths were applied to each cell in order to minimise the divergence of the scan head movements from the set path due to its inertia. The scanning order of the identified cells was optimised using the travelling salesman method.

### 4.4. Data Processing

Data evaluation was performed using custom MATLAB scripts available at http://downloadables.ttk.hu/heja/IJMS_2024/ (accessed on 9 August 2024). Classification of cells as either astrocytes or neurons was performed based on OGB-1 and SR101 fluorescence within an ROI according to the following automated protocol: (1) ROIs with sizes smaller than 100 pixels (approximately 120 µm^2^) were not considered cells and were excluded from further analysis; (2) fluorescence intensity within the ROI relative to a 7-pixel-wide ring around the ROI was calculated for both OGB-1 and SR-101 channels; (3) a cell was classified as an astrocyte if the above relative SR-101 intensity was higher than 2 or if it was higher than 1.3 and the eccentricity of the ROI was smaller than 0.85; (4) a cell was classified as a neuron if the SR-101 intensity was smaller than 0.65× (OGB-1 intensity) and the relative OGB-1 intensity calculated in step 2 was higher than 2 or if it was higher than 1.3 and the eccentricity of the ROI was smaller than 0.85. The cell selection criteria were intentionally set to be strict in order to exclude areas that represented non-somatic signals.

ΔF/F_0_ traces were calculated from raw fluorescent intensity data and were subjected to wavelet analysis using the complex Morlet wavelet with a bandwidth of 1 and centre frequencies of 2 and 6 Hz for delta- and theta-band wavelets, respectively.

Phase-locking values (PLVs) were calculated by first filtering the ΔF/F_0_ traces of all cells using a finite impulse response (FIR) filter with an order of 1000/stopband frequency/sampling frequency, then applying the following formula:$$PLV=\frac{1}{N}\left|\sum_{i=1}^{N}e^{1i*(\phi 1-\phi 2)}\right|,$$
where Φ_1_ and Φ_2_ are the phases of the Hilbert transform of the two filtered ΔF/F_0_ traces to be compared, and N is the number of data points of the ΔF/F_0_ trace.

All data are expressed as means ± SEM and were analysed using one-way analysis of variance (ANOVA) with Bonferroni post hoc tests (OriginPro 2023b). A value of *p* < 0.05 was considered significant.

## Figures and Tables

**Figure 1 ijms-25-08911-f001:**
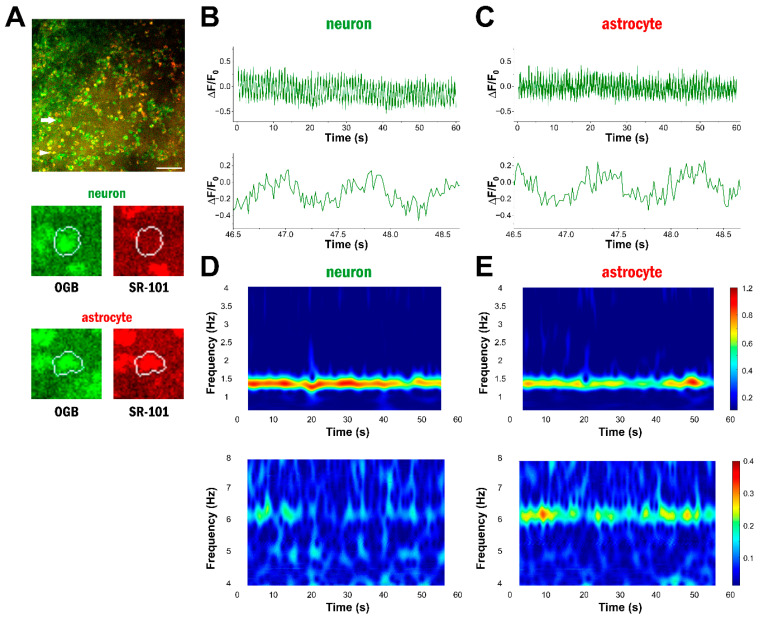
Astrocytes display high-frequency Ca^2+^ oscillations. (**A**) OGB-1 and SR-101 labelling in layer 2 of the V1 cortex in vivo during ketamine/xylazine anaesthesia. The analysed neuron is marked by an arrow, and the analysed astrocyte is marked by an arrowhead. Boundary of the ROI in which the cell type was determined is shown in white. Scale bar: 100 µm. (**B**) OGB-1 fluorescence intensity changes in the selected neuron during the 60 s long imaging session (top) and a close-up view of the ΔF/F_0_ trace showing high-frequency changes (bottom). (**C**) OGB-1 fluorescence intensity changes in the selected astrocyte during the 60 s long imaging session (top) and a close-up view of the ΔF/F_0_ trace showing high-frequency changes (bottom). (**D**) Wavelet analysis of the selected neuron in the delta (0.5–4 Hz, top) and the theta (4–8 Hz, bottom) frequency ranges. (**E**) Wavelet analysis of the selected astrocyte in the delta (0.5–4 Hz, top) and the theta (4–8 Hz, bottom) frequency ranges.

**Figure 2 ijms-25-08911-f002:**
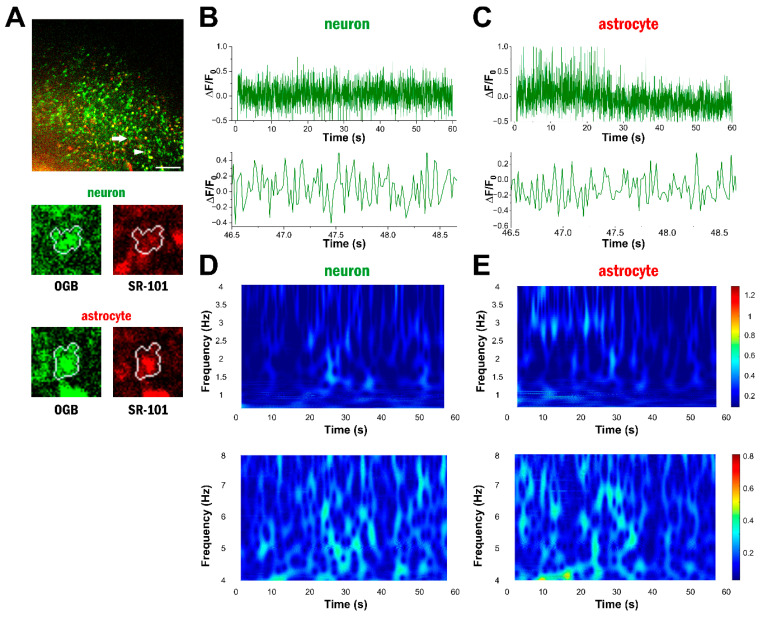
High-frequency Ca^2+^ oscillations do not emerge in fentanyl anaesthesia. (**A**) OGB-1 and SR-101 labelling in layer 2 of the V1 cortex in vivo during fentanyl anaesthesia. The analysed neuron is marked by an arrow, and the analysed astrocyte is marked by an arrowhead. Boundary of the ROI in which the cell type was determined is shown in white. Scale bar: 100 µm. (**B**) OGB-1 fluorescence intensity changes in the selected neuron during the 60 s long imaging session (top) and a close-up view of the ΔF/F_0_ trace (bottom). (**C**) OGB-1 fluorescence intensity changes in the selected astrocyte during the 60 s long imaging session (top) and a close-up view of the ΔF/F_0_ trace (bottom). (**D**) Wavelet analysis of the selected neuron in the delta (0.5–4 Hz, top) and the theta (4–8 Hz, bottom) frequency ranges. (**E**) Wavelet analysis of the selected astrocyte in the delta (0.5–4 Hz, top) and the theta (4–8 Hz, bottom) frequency ranges.

**Figure 3 ijms-25-08911-f003:**
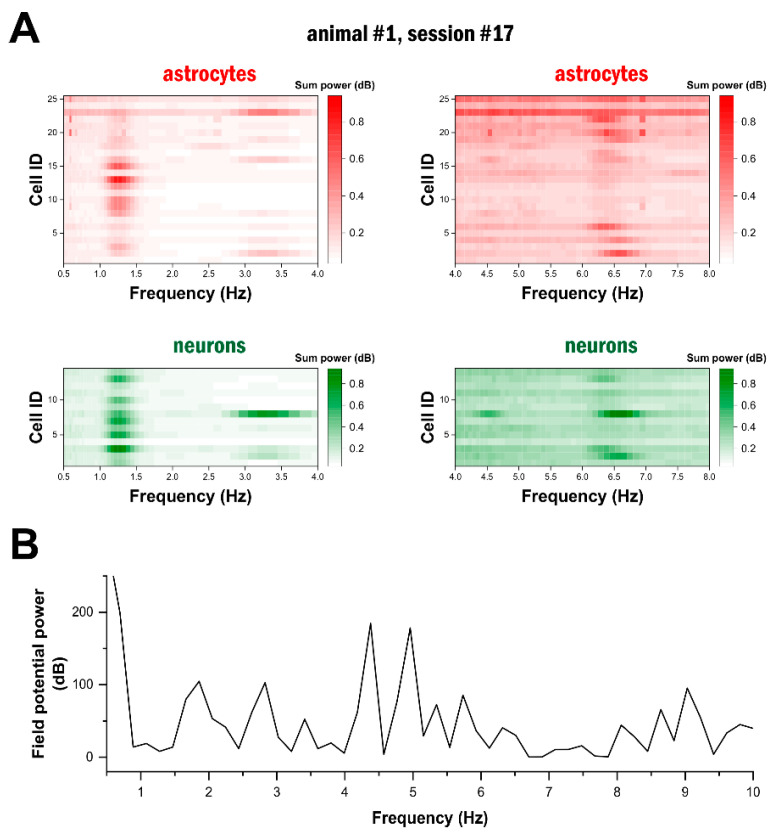
Delta- and theta-band activity is widespread in the astrocytic network. (**A**) Spectral resolution of the Ca^2+^ signals of all identified astrocytes (top) and neurons (bottom) in a 60 s imaging session in animal #1 in the delta (0.5–4 Hz, left) and the theta (4–8 Hz, right) frequency bands. Each horizontal line represents the frequency distribution of a single cell. (**B**) Power spectrum of the simultaneously measured electrophysiological field potential signal in the same imaging session as in (**A**). (**C**) Spectral resolution of the Ca^2+^ signals of all identified astrocytes (top) and neurons (bottom) in a 60 s imaging session in animal #2 in the delta (0.5–4 Hz, left) and the theta (4–8 Hz, right) frequency bands. Each horizontal line represents the frequency distribution of a single cell. (**D**) Power spectrum of the simultaneously measured electrophysiological field potential signal in the same imaging session as in (**C**).

**Figure 4 ijms-25-08911-f004:**
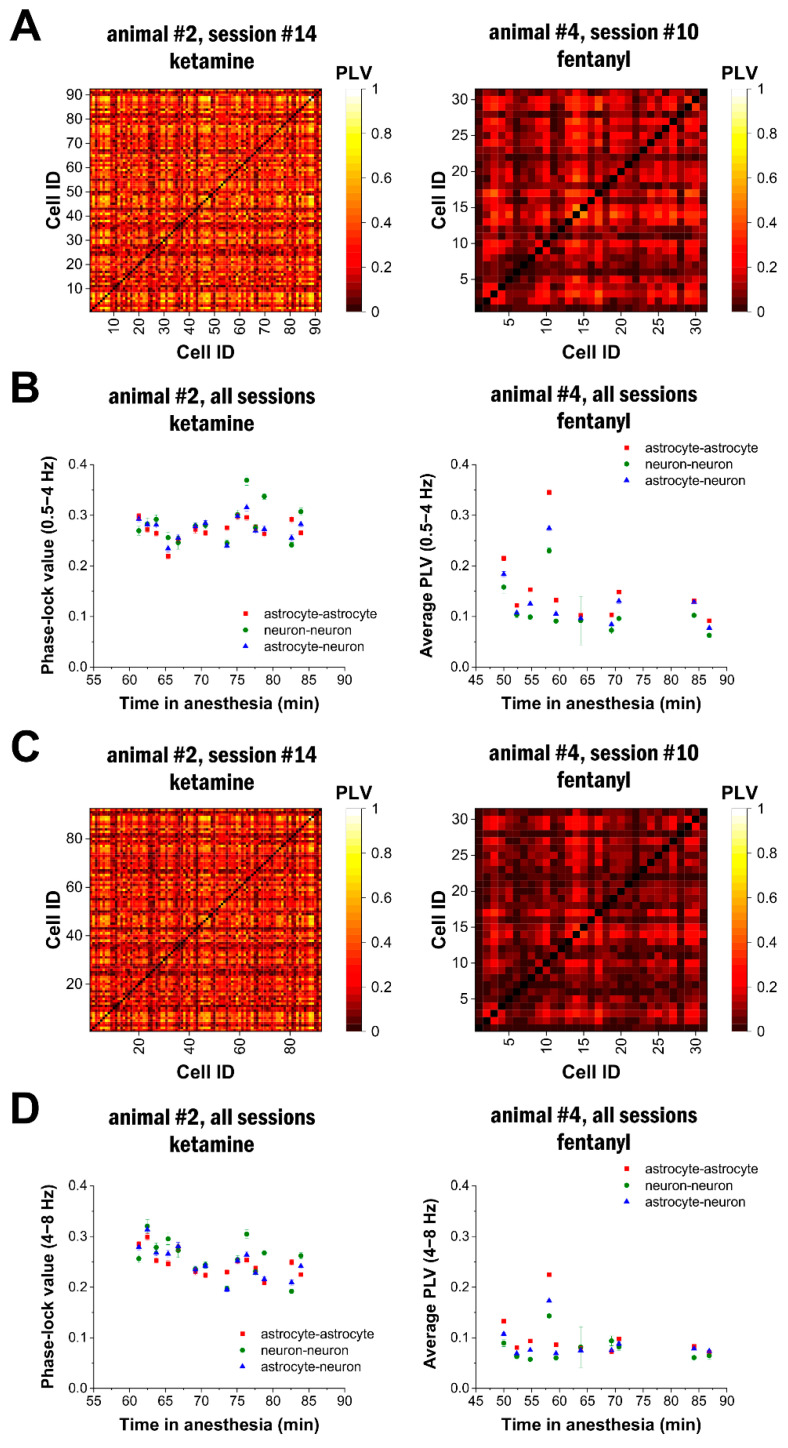
Long-term appearance of delta- and theta-band activity in the astrocytic network. (**A**) Phase-locking value (PLV) between Ca^2+^ traces of all cell pairs in a 60 s long imaging session in the delta frequency range (0.5–4 Hz) in ketamine–xylazine (left) or fentanyl (right) anaesthesia. (**B**) Average PLV in all imaging sessions in an animal, calculated separately for astrocyte–astrocyte, astrocyte–neuron, and neuron–neuron pairs in the delta frequency range (0.5–4 Hz) in ketamine–xylazine (left) or fentanyl (right) anaesthesia. (**C**) Phase-locking value (PLV) between Ca^2+^ traces of all cell pairs in a 60 s long imaging session in the theta frequency range (4–8 Hz) in ketamine–xylazine (left) or fentanyl (right) anaesthesia. (**D**) Average PLV in all imaging sessions in an animal, calculated separately for astrocyte–astrocyte, astrocyte–neuron, and neuron–neuron pairs in the theta frequency range (4–8 Hz) in ketamine–xylazine (left) or fentanyl (right) anaesthesia.

**Figure 5 ijms-25-08911-f005:**
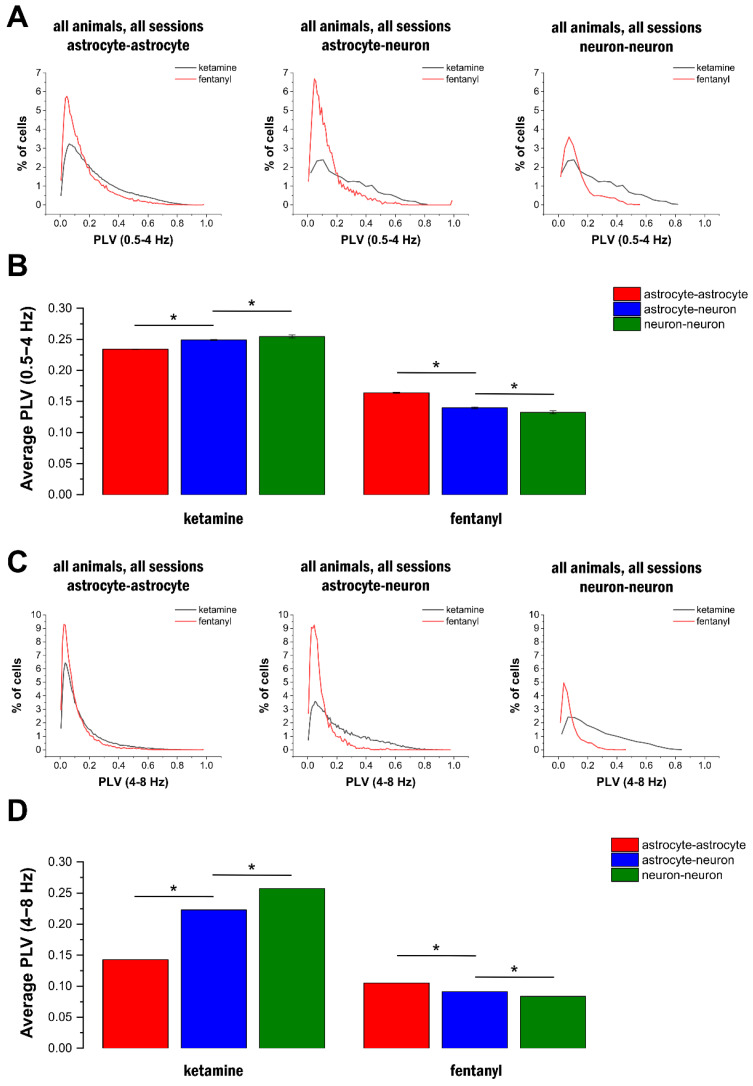
Highly synchronised activity develops in ketamine–xylazine anaesthesia. (**A**) Distribution of delta range phase-lock values (PLVs) for all investigated astrocyte–astrocyte (left), astrocyte–neuron (middle), and neuron–neuron (right) cell pairs in ketamine–xylazine and fentanyl anaesthesia. (**B**) Average delta range PLVs for all investigated astrocyte–astrocyte, astrocyte–neuron, and neuron–neuron cell pairs in ketamine–xylazine and fentanyl anaesthesia. (**C**) Distribution of theta range PLVs for all investigated astrocyte–astrocyte (left), astrocyte–neuron (middle), and neuron–neuron (right) cell pairs in ketamine–xylazine and fentanyl anaesthesia. (**D**) Average theta range PLVs for all investigated astrocyte–astrocyte, astrocyte–neuron, and neuron–neuron cell pairs in ketamine–xylazine and fentanyl anaesthesia. Asterisks mean significant differences (*p* < 0.05).

## Data Availability

Raw data and evaluation scripts are publicly available at http://downloadables.ttk.hu/heja/IJMS_2024/ (accessed on 9 August 2024).

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
