# Peer review of "High-Frequency Imaging Reveals Synchronised Delta- and Theta-Band Ca2+ Oscillations in the Astrocytic Soma In Vivo"

_ijms, 2024, doi:10.3390/ijms25168911_

Round 1
Reviewer 1 Report
Comments and Suggestions for Authors
The manuscript explores a novel and interesting aspect of astrocytic activity by employing high-frequency two-photon imaging to reveal fast Ca2+ oscillations in the delta (0.5-4 Hz) and theta (4-8 Hz) frequency bands in astrocytic soma. This is a significant departure from the traditional view that astrocytes are slow responders in terms of Ca2+ signaling. However, despite the innovative approach and potential implications, several critical issues need to be addressed.
Comments:
1. The manuscript lacks sufficient control experiments to rule out potential artifacts, such as those caused by anesthesia. The authors should include additional control groups or conditions to validate their findings beyond ketamine-xylazine anesthesia.
2. The methodology for identifying astrocytes using SR-101 and confirming OGB-1 loading should be more rigorously described and validated with appropriate controls.
3. While the presence of delta and theta oscillations in astrocytic Ca2+ signals is intriguing, the manuscript does not adequately address the potential sources of these signals. More discussion is needed on whether these oscillations are intrinsic to astrocytes or if they are influenced by neighboring neuronal activity.
4. The statistical methods used to analyze the imaging data are not thoroughly described. Details on the statistical tests, including how they were applied and the rationale for their selection, should be included.
5. Some figures, particularly those showing wavelet analysis and spectral resolution, are difficult to interpret. Improved clarity in the presentation of these figures is necessary, along with more detailed legends that explain the data comprehensively.
6. The authors should provide a detailed description of the data acquisition and processing protocols to ensure reproducibility. This includes sharing MATLAB scripts or any other software used in the analysis.
7. The manuscript would benefit from more mechanistic insights into how the observed Ca2+ oscillations are generated and propagated within astrocytes. Potential molecular mechanisms and pathways should be explored or hypothesized based on existing literature.
Author Response
The manuscript explores a novel and interesting aspect of astrocytic activity by employing high-frequency two-photon imaging to reveal fast Ca2+ oscillations in the delta (0.5-4 Hz) and theta (4-8 Hz) frequency bands in astrocytic soma. This is a significant departure from the traditional view that astrocytes are slow responders in terms of Ca2+ signaling. However, despite the innovative approach and potential implications, several critical issues need to be addressed.
We thank the reviewer for his/her valuable comments and criticism. In the revised version of the manuscript, we provide a substantial amount of new experimental data to address his/her concerns. Data from 3 animals anesthetized by fentanyl was added to the revised version of the manuscript together with data from an additional animal anesthetized by ketamine-xylazine. Moreover, the manuscript was extended with further details of the experimental results as well as additional discussions.
All the raw data and the MATLAB scripts used to evaluate the results are made publicly available.
Comments:
- The manuscript lacks sufficient control experiments to rule out potential artifacts, such as those caused by anesthesia. The authors should include additional control groups or conditions to validate their findings beyond ketamine-xylazine anesthesia.
In accordance with the reviewer’s suggestion, we tested whether high-frequency astrocytic oscillations are present in another type of anesthesia, induced by fentanyl + midazolam + medetomidine. In contrast to ketamine anesthesia that is known to induce permanent slow-wave activity, fentanyl does not trigger permanent slow-wave sleep. Correspondingly, we observed that a lower level of synchronization emerged in both the astrocytic and the neuronal network compared to the ketamine-xylazine anesthesia. The decreased synchronization was observed in both the delta and theta range. Furthermore, we also revealed that phase coupling between neurons was higher than between astrocytes in the ketamine-xylazine anesthesia. On the contrary, astrocytic activity was more phase coupled than that of neurons in the fentanyl anaesthesia. In our view, the significant difference between anesthesia types that do or do not induce synchronized network activity rules out motion artefacts as a potential source of the observed signals.
- The methodology for identifying astrocytes using SR-101 and confirming OGB-1 loading should be more rigorously described and validated with appropriate controls.
According to the reviewer’s suggestion, we have provided a detailed description of the cell type classification protocol (page 11-12):
“Data evaluation was performed using custom Matlab scripts available at http://downloadables.ttk.hu/heja/IJMS_2024/. Classification of cells as either astrocytes or neurons were done based on OGB-1 and SR101 fluorescence within a ROI according to the following automated protocol: 1) ROIs with size smaller than 100 pixel (approximately 120 µm2) were not considered as cells and were excluded from further analysis; 2) fluorescence intensity within the ROI relative to a 7 pixel wide ring around the ROI was calculated for both OGB-1 and SR-101 channels; 3) a cell was classified as astrocyte if the above relative SR-101 intensity was higher than 2 or it was higher than 1.3 and the eccentricity of the ROI was smaller than 0.85; 4) a cell was classified as a neuron if the SR-101 intensity was smaller than 0.65 x (OGB-1 intensity) and the relative OGB-1 intensity calculated in step 2 was higher than 2 or it was higher than 1.3 and the eccentricity of the ROI was smaller than 0.85. The cell selection criteria were intentionally set to be strict in order to exclude areas that represent non-somatic signals.”
We have also deposited all the Matlab scripts used for data analysis, together with the original raw data at http://downloadables.ttk.hu/heja/IJMS_2024/
- While the presence of delta and theta oscillations in astrocytic Ca2+ signals is intriguing, the manuscript does not adequately address the potential sources of these signals. More discussion is needed on whether these oscillations are intrinsic to astrocytes or if they are influenced by neighboring neuronal activity.
We agree with the reviewer that identification of the molecular mechanisms responsible for generation and spreading of the oscillatory astrocytic activity is of crucial importance both for better understanding the signal processing and to provide potential new astrocytic targets for drug development. It is to mention that our original purpose with the development of high-frequency astrocytic calcium imaging was to investigate the role of gap junctions in astrocytic calcium signalling. Astrocytic connexins indeed play a major role in the generation of recurrent activity on the astrocytic network level. However, the complexity of this issue necessitated the presentation of the gap junction-related experimental data in a separate publication.
In accordance with the reviewer’s advice, however, in the revised version of the manuscript we added a detailed discussion about the potential sources of astrocytic calcium fluctuations and how this cell-level activity can be emerged on the network level (page 8-9):
“Although our current goal was merely to reveal the existence of high-frequency oscillations in astrocytic soma, the appearance and widespread presence of delta and theta band activities in astrocytes raises the question whether these oscillations originate from the astrocytes themselves or they are mirroring the already established activity pattern of the neuronal network. Autonomous signals within the astrocytic syncytium can arise as a result of several mechanisms and sources. Astrocytic microdomains are capable of generating autonomic Ca2+ oscillations via the interplay between Na+ dependent neurotransmitter transporters, such as GAT3 [30] or EAAT2 [31] and NCX. The influx of Na+ can reverse the operation of NCX, thereby resulting in the uptake of Ca2+ [32]. This has been confirmed by our previous in silico simulations [22]. In the astrocytic soma and main processes the release of Ca2+ from intracellular stores can contribute to the development of autonomous Ca2+ signals, through the activation of GPCRs (like InsP3R2), and the subsequent activation of IP3 dependent release mechanisms [20,33]. Even in the absence of these mechanisms, spontaneous Ca2+ signals can still arise as a result of mitochondrial activity, specifically the activation of permeability transition pore (mPTP) [20,33]. The depletion of Ca2+ stores can also trigger Ca2+ entry trough STIMs and Orai channels [33,34].
Alternatively, the signals may propagate from the synapses, passed on to astrocytic microdomains, using extracellular Ca2+ sources [18,19]. If the synchronized neuronal activity is strong enough to activate a significant number of astrocytic microdomains, the signal could spread to the soma through the main processes, where intracellular Ca2+ stores are used for further propagation [12,24,35]. From here the signals can spread through the astrocytic syncytium through gap junctions, explaining the network-wide activity in this cell population. Previous experimental data, however, suggest that Ca2+ signalling between neurons and astrocytes may not be unidirectional. Under similar circumstances, synchronized slow wave activity of astrocytes developed earlier than that of neurons [5]. Furthermore, specific blockade of astrocytic gap junctions resulted in a significant reduction of neuronal synchronizations in the slow wave activity frequency range (0.5-2 Hz) [5] and also diminished high frequency epileptic activity [36]. In vitro data also demonstrate that specific inhibition or enhancement of astrocytic Ca2+ activity can decrease or increase neuronal synchronization, respectively [11]. Further studies will be needed to discriminate between neuronal and inherent astrocytic activity as the source of the fast astrocytic oscillations”
- The statistical methods used to analyze the imaging data are not thoroughly described. Details on the statistical tests, including how they were applied and the rationale for their selection, should be included.
We have updated the Methods section to include the statistical test used. Also, we have made available all the MATLAB scripts used to analyze the data, together with the raw data themselves.
- Some figures, particularly those showing wavelet analysis and spectral resolution, are difficult to interpret. Improved clarity in the presentation of these figures is necessary, along with more detailed legends that explain the data comprehensively.
More detailed explanation about the content of wavelet analysis, especially those shown on Fig3 is provided in the Results section.
- The authors should provide a detailed description of the data acquisition and processing protocols to ensure reproducibility. This includes sharing MATLAB scripts or any other software used in the analysis.
We have made available all the MATLAB scripts used to analyze the data, together with the raw data themselves.
- The manuscript would benefit from more mechanistic insights into how the observed Ca2+ oscillations are generated and propagated within astrocytes. Potential molecular mechanisms and pathways should be explored or hypothesized based on existing literature.
As described in our response to point 3, we added a detailed discussion about both the potential sources of astrocytic calcium fluctuations and how this cell-level activity can be propagated in the astrocytic network (page 8-9).
Reviewer 2 Report
Comments and Suggestions for Authors
In this study, Péter and Héja utilized high-frequency two-photon imaging to identify fast Ca2+ oscillations in delta (0.5-4 Hz) and theta (4-8 Hz) frequencies in the soma of astrocytes in rat cortex under ketamine-thiophene anesthesia. These signals were observed in a proportion of astrocytes and exhibited synchronization within the astrocyte network. While the findings are novel, additional experimental verification is crucial to strengthen the evidence. The following aspects need to be investigated or resolved:
1. This study used two rats for experiments. An increase in the sample size is necessary to enhance the robustness of the findings and the accuracy of the conclusions.
2. The quality of Fig1A could be higher, especially the enlarged picture. The fluorescence intensity of OGB selected by the author to represent neurons is low, the fluorescence intensity of SR-101 selected to represent astrocytes is low, and astrocyte processes cannot be seen. Why not choose cells with more obvious green fluorescence and more obvious yellow fluorescence? Moreover, it is necessary to detect Ca2+ oscillations in astrocyte processes and compare these findings with the frequency bands in the soma. This will make the results more credible and solid.
3. Is the Ca2+ oscillation frequency band of astrocytes observed in this study limited to rats under ketamine-xylazine anesthesia? How to prove it? It is crucial to add background literature to this part for discussion.
4. Many font and formatting errors in the text require review and correction.
Author Response
In this study, Péter and Héja utilized high-frequency two-photon imaging to identify fast Ca2+ oscillations in delta (0.5-4 Hz) and theta (4-8 Hz) frequencies in the soma of astrocytes in rat cortex under ketamine-thiophene anesthesia. These signals were observed in a proportion of astrocytes and exhibited synchronization within the astrocyte network. While the findings are novel, additional experimental verification is crucial to strengthen the evidence. The following aspects need to be investigated or resolved:
We thank the reviewer for his/her valuable comments and criticism. In the revised version of the manuscript, we provide a substantial amount of new experimental data to address his/her concerns. Data from 3 animals anesthetized by fentanyl was added to the revised version of the manuscript together with data from an additional animal anesthetized by ketamine-xylazine. Moreover, the manuscript was extended by further details of the experimental results as well as additional discussions thereof.
All the raw data and the MATLAB scripts used to evaluate the results are made publicly available.
- This study used two rats for experiments. An increase in the sample size is necessary to enhance the robustness of the findings and the accuracy of the conclusions.
In accordance with the reviewer’s suggestion, we included data from 4 new animals. One of them was anesthetized by ketamine/xylazine and 3 were anesthetized by fentanyl. In addition to the new data, in the revised manuscript we provide further analysis of the imaging results.
- The quality of Fig1A could be higher, especially the enlarged picture. The fluorescence intensity of OGB selected by the author to represent neurons is low, the fluorescence intensity of SR-101 selected to represent astrocytes is low, and astrocyte processes cannot be seen. Why not choose cells with more obvious green fluorescence and more obvious yellow fluorescence? Moreover, it is necessary to detect Ca2+ oscillations in astrocyte processes and compare these findings with the frequency bands in the soma. This will make the results more credible and solid.
According to the reviewer’s suggestion, we replaced the representative astrocyte and neuron on Fig1.
We fully agree with the Reviewer that after revealing the presence of high-frequency Ca2+ oscillations on the astrocytic soma, it will be of crucial importance to evaluate the connection between somatic signals and the corresponding Ca2+ fluctuations in the processes. Unfortunately, however, the labeling method we used does not allow measuring Ca2+ in the processes. We used an organic Ca2+ sensitive dye OGB-1 that does not label the processes. It is to mention that our original purpose with the development of high-frequency astrocytic calcium imaging was to investigate the role of gap junctions in astrocytic calcium signalling and memory formation (due to the complexity of this issue, these data is about to be published in a separate publication). According to this goal, we have chosen a labeling method that can be immediately applied, because the cognitive tests are already rather time consuming. Therefore, we avoided using viral delivery of genetic Ca2+ sensors. Furthermore, we opted to use rats instead of mice due to their superior performance in cognitive tasks. Unfortunately, however, genetic tools are significantly more limited in rats than in mice. Another reason to use an organic dye was that we experienced that more widespread labeling can be achieved in the field of view with OGB-1 than with genetic calcium sensors. Therefore, different experimental sets need to be used in further works to explore the relationship between astrocytic Ca2+ oscillations in different compartments.
- Is the Ca2+ oscillation frequency band of astrocytes observed in this study limited to rats under ketamine-xylazine anesthesia? How to prove it? It is crucial to add background literature to this part for discussion.
We thank the Reviewer for raising this issue. In the revised version of the manuscript, we evaluated the high-frequency astrocytic Ca2+ signals in fentanyl anesthesia, which (in contrast to ketamine/xylazine) does not induce permanent slow wave activity. Indeed, we found that both delta and theta range activity was significantly reduced in fentanyl anesthesia. This observation confirms that the measured signals are not initiated by motion artefacts. The new results are described on page 5:
“The appearance of delta- and theta range activity in astrocytes and neurons is most likely induced by ketamine/xylazine anaesthesia that generates permanent slow-wave sleep in rats [26,27]. To further confirm that the observed high-frequency astrocytic signals correspond to the induced slow-wave activity and are not caused by motion artefacts, we investigated the appearance of high frequency Ca2+ signals in 3 rats anaesthetized by fentanyl (25 µg/kg), medetomidine (0.25 mg/kg) and midazolam (2 mg/kg). In these 84 imaging sessions, 7-70 cells (28 ± 2 cells in average) were recorded at 29-101 Hz sampling frequencies, 95 ± 1 % of them were identified as astrocytes. In fentanyl anesthesia, most of the cells did not show delta- or theta band Ca2+ oscillations (Fig. 2), confirming that the high-frequency astrocytic Ca2+ signals, observed during ketamine/xylazine anesthesia were not due to motion artefacts triggered by breathing, heartbeat or movement instability of the scanning head.”
Comparison of high-frequency Ca2+ signals between ketamine/xylazine and fentanyl anesthesia is further analyzed in Figures 2, 4 and 5.
- Many font and formatting errors in the text require review and correction.
We have extensively reviewed the text and corrected typographical and formatting issues.
Reviewer 3 Report
Comments and Suggestions for Authors
The manuscript reports results on oscillatory astrocytic activity delta and theta frequency ranges observed in calcium signal using two-photon imaging.
I find the topic very interesting and of great scientific relevance, as the importance of glial cells to brain computation has been generally overlooked. The methods are sound, conclusions are reasonably constrained within the limits allowed by results, and the text is well written.
There are a few issues that authors must address before the paper is suitable for publication.
MAJOR ISSUES
My biggest concern is that conclusions are drawn from observations obtained from only two animals. Even though this is more a “proof of principle” (astrocytes display delta and theta oscillatory activity) kind of paper, which does not require any kind of statistical comparison, I found a sample number of only two too small. Is this common? Are there any kind of guidelines for such reporting? Please, answer this and, eventually, include clarifications in this sense in the body of the manuscript.
ISSUE ON FORMAT
IJMS does not adopt the classical IMRAD format for the sequence of sections, leaving methods to the end of the text instead. I think this creates problems, as authors tend to include some methodology in the beginning of the results section for clarity. I think it was the case here and I have the feeling that they included too many details. Please, review this and consider moving some of this content to the proper section.
OTHER ISSUES
RESULTS
Although they are not the same thing, of course, it is expected that frequencies found in the calcium signals would highly coincide with those of electrophysiology, wouldn’t they? It does not seem to be the case, particularly in figure 2A / 2B. What do authors find about this?
DISCUSSION
What are the findings using astrocyte electrophysiology that may corroborate present findings?
Authors state: “All signals were present and synchronized on the network level in both astrocytes and neurons”. What objective result supports this conclusion? The average maximal cross correlation? Considering the big number of cells (around 100) and thus the big number of possible pair-wise correlations, there will always be a bigger number. Moreover, what configures a strong correlation? What is the threshold for high or low correlation? In comparison to what? I find this statement somewhat subjective.
METHODS
Why did authors choose to use female instead of male, which is the conventional choice?
Please provide additional information on electrophysiology, including electrode type/material/dimensions, recording parameters, and target areas.
Please, include references to the state-of-the-art literature on fast calcium imaging.
Was there any kind of supplementation of anesthesia during the process? Why are there some gaps in the results time-points (e.g. around 70 min for animal 1)?
Comments on the Quality of English Language
Authors write in good English and there are no major issues in this aspect. Yet, a few typos here and there still exist and I suggest one additional careful review in this sense.
Author Response
The manuscript reports results on oscillatory astrocytic activity delta and theta frequency ranges observed in calcium signal using two-photon imaging.
I find the topic very interesting and of great scientific relevance, as the importance of glial cells to brain computation has been generally overlooked. The methods are sound, conclusions are reasonably constrained within the limits allowed by results, and the text is well written.
There are a few issues that authors must address before the paper is suitable for publication.
MAJOR ISSUES
My biggest concern is that conclusions are drawn from observations obtained from only two animals. Even though this is more a “proof of principle” (astrocytes display delta and theta oscillatory activity) kind of paper, which does not require any kind of statistical comparison, I found a sample number of only two too small. Is this common? Are there any kind of guidelines for such reporting? Please, answer this and, eventually, include clarifications in this sense in the body of the manuscript.
In accordance with the reviewer’s suggestion, we included data from 4 new animals. One of them was anesthetized by ketamine/xylazine and 3 were anesthetized by fentanyl. In addition to the new data, in the revised manuscript we provide further analysis of the imaging results.
ISSUE ON FORMAT
IJMS does not adopt the classical IMRAD format for the sequence of sections, leaving methods to the end of the text instead. I think this creates problems, as authors tend to include some methodology in the beginning of the results section for clarity. I think it was the case here and I have the feeling that they included too many details. Please, review this and consider moving some of this content to the proper section.
Although we agree with the Reviewer that placing the Methods section at the end of the manuscript may lead to duplication of some experimental or data evaluation details, issues arose by the other Reviewers prompted us to specify these details in the Results section as well.
OTHER ISSUES
RESULTS
Although they are not the same thing, of course, it is expected that frequencies found in the calcium signals would highly coincide with those of electrophysiology, wouldn’t they? It does not seem to be the case, particularly in figure 2A / 2B. What do authors find about this?
We agree with the Reviewer that comparing network activities of astrocytes and neurons will be of crucial importance in understanding high-frequency astrocytic oscillations. Regarding the frequency coupling between astrocytic and neuronal rhythms: it probably strongly depends on the distance between the measuring electrode and the field of view used for the imaging measurements. Since the location of the electrode was fixed, while the field of view was not (our main goal in selecting the field of view was to find a sufficiently high number of labelled astrocytes), the distance between the two sites was variable. Sometimes we found close matching between the identified astrocytic and neuronal characteristic frequencies, but sometimes the correlation was weaker. It is also to mention that the exact frequency maximum identified by the wavelet analysis may vary by 0.5-1 Hz by changing the wavelet type or the wavelet parameters. In a broader sense, astrocytic signals may require a different set of parameters to describe their functions. Currently, we applied terms and methodology developed for neurons, for example by defining the frequency ranges of delta and theta bands. However, simply mirroring the electrophysiological terminology may not necessarily be the best approach to understand astrocyte physiology.
DISCUSSION
What are the findings using astrocyte electrophysiology that may corroborate present findings?
Isopotentiality in the astrocytic network is mainly maintained by gap junctions. We previously found that blockade of gap junctions also disrupts network-level Ca2+ oscillations in astrocytes. Therefore, astrocyte electrophysiology and Ca2+ signalling may be correlated. However, since we did not measure astrocyte membrane potentials in the current study, our results do not allow making conclusions.
Authors state: “All signals were present and synchronized on the network level in both astrocytes and neurons”. What objective result supports this conclusion? The average maximal cross correlation? Considering the big number of cells (around 100) and thus the big number of possible pair-wise correlations, there will always be a bigger number. Moreover, what configures a strong correlation? What is the threshold for high or low correlation? In comparison to what? I find this statement somewhat subjective.
We used the above evaluation due to its simplicity and its easily comprehensible correlation with the level of synchronization in the network. However, we fully agree with the Reviewer that it is preferred to use more objectively calculated parameters. In the revised version of the manuscript, we fully removed our previous evaluation and instead calculated the phase-lock value to quantify network synchronization. This standard parameter, widely used in the electrophysiological literature, allowed us to quantify the synchrony of the Ca2+ signals and also to compare data from animals anaesthetized with ketamine/xylazine or fentanyl.
METHODS
Why did authors choose to use female instead of male, which is the conventional choice?
The initial measurements, where we discovered the high-frequency oscillations, were part of a series of experiments investigating the effects of astrocytic gap junctions on slow wave activity. Because of this, extra equipment needed to be fitted to the skull. Females were chosen because their skulls are more accessible, and the available area for fitting equipment is slightly wider. As the surgical protocol developed further, we eventually transitioned into using males instead, when we substituted the aluminum plate used for head fixation with a stereotaxic frame, thereby eliminating the need for extra space on the skull. The newly included animals undergoing fentanyl anesthesia were all males.
Please provide additional information on electrophysiology, including electrode type/material/dimensions, recording parameters, and target areas.
The steel legs of a JST-XH Male 2 Pin Connector were used as electrodes. The legs were spread 7.5 mms apart from each other and secured into holes drilled through the skull. We adapted this connector because it enables a secure and stable connection even in freely moving animals. The downside is that it is fairly bulky, and therefore had to be fitted above the contralateral hemisphere from the cranial window to allow enough space for the microscope objective during measurements. The first hole was drilled at the edge of the bregma through the parietal bone on the very edge of the exposed part of the skull, the second one 7.5 mms dorsally, towards the middle of the parietal bone.
Please, include references to the state-of-the-art literature on fast calcium imaging.
In accordance with the reviewer’s suggestion, we added references to current state-of-art fast imaging studies to the Introduction section.
Was there any kind of supplementation of anesthesia during the process? Why are there some gaps in the results time-points (e.g. around 70 min for animal 1)?
Ketamine anesthesia had to be supplemented occasionally during measurements in some animals. The gap in the results of animal #1 is an example of this. Other gaps are usually a result of technical difficulties, as certain pieces of software and hardware needed to be occasionally rebooted, or the field of view needed to be reacquired.
Round 2
Reviewer 1 Report
Comments and Suggestions for Authors
None
Reviewer 2 Report
Comments and Suggestions for Authors
The Authors have addressed all of my concerns.